# Sparse Autoencoders Trained on the Same Data Learn Different Features

**Gonçalo Paulo, Nora Belrose**
EleutherAI
{goncalo,nora}@eleuther.ai

## Abstract

Sparse autoencoders (SAEs) are a useful tool for uncovering human-interpretable features in the activations of large language models (LLMs). While some expect SAEs to find the true underlying features used by a model, our research shows that SAEs trained on the same model and data, differing only in the random seed used to initialize their weights, identify different sets of features. For example, in an SAE with 131K latents trained on a feedforward network in Llama 3 8B, only 30% of the features were shared across different seeds. We observed this phenomenon across multiple layers of three different LLMs, two datasets, and several SAE architectures. While ReLU SAEs trained with the L1 sparsity loss showed greater stability across seeds, SAEs using the state-of-the-art TopK activation function were more seed-dependent, even when controlling for the level of sparsity. Our results suggest that the set of features uncovered by an SAE should be viewed as a pragmatically useful decomposition of activation space, rather than an exhaustive and universal list of features "truly used" by the model.

## 1 Introduction

Sparse autoencoders (SAEs) are an interpretability tool used to decompose neural network activations into human-understandable features (Cunningham et al., 2023). They address the problem of *polysemanticity*, where individual neurons activate in semantically diverse contexts, defying any simple explanation (Arora et al., 2018; Elhage et al., 2022). SAEs consist of two parts: an encoder that transforms activation vectors into a sparse, higher-dimensional latent space, and a decoder that projects the latents back into the original space. Both parts are trained jointly to minimize reconstruction error. Recently, SAEs have been scaled to state-of-the-art large language models, like GPT-4 (Gao et al., 2024) and Claude 3 Sonnet (Templeton et al., 2024).

Many researchers hope SAEs can be used to "identify and enumerate over all features in a model" (Elhage et al., 2022), which might allow us to check certain safety properties, such as that "a model will never lie" (Olah, 2023). These hopes seem to presuppose that there is a unique, objective decomposition of a neural network into features, and that SAEs can uncover this decomposition (Smith, 2024). In this paper, we test this presupposition by measuring the degree to which SAE features depend on the random seed used to initialize their weights.

It is somewhat nontrivial to compare features learned by different SAEs, since the latents of the SAE have no inherent ordering.[1] Given a trained SAE $\mathcal{M}$, we can generate a "shuffled" SAE $\mathcal{M}'$ by randomly permuting the rows of $\mathcal{M}$'s encoder matrix, and rearranging the columns of its decoder matrix using the inverse permutation.[2] $\mathcal{M}$ and $\mathcal{M}'$ represent the same function and contain *the same features*, up to an irrelevant permutation symmetry, and yet their weights may look very different.

---

[1] In this work we will use "latent" to refer to a row in an SAE encoder matrix and its corresponding row in the decoder matrix. By contrast, "feature" is reserved for the concept that a latent may represent. We assume that if two latents are matched by our method, they do refer to *some* shared concept worth calling a "shared feature," although it may not make sense to a human. It would be a misnomer to speak of "shared latents."

[2] This is generally true for any MLP with elementwise nonlinearities. Indeed, for ReLU networks there is also a continuous symmetry: the function represented by the model is unchanged when the pre-activation is scaled by $s$ and the post-activation is scaled by $s^{-1}$. The standard training recipe for SAEs eliminates this symmetry, however, by constraining decoder vectors to be unit norm (Bricken et al., 2023).

This means that if independently trained SAEs with different random initializations do learn the same features, we would still expect their latents to be arranged in different orders, and hence we cannot directly compare them.

We can get around this problem by computing a bijective *matching* of each latent in the first SAE with a unique counterpart latent in second SAE. This matching should be optimal in the sense that it maximizes the average "similarity" between the matched latents. This ensures that, in the case where one SAE has actually been generated by shuffling the latents of another, we will conclude that the resulting SAEs do indeed have the same features. Luckily, the Hungarian algorithm is known to efficiently compute this optimal matching, and has been used to align independently trained networks before (Ainsworth et al., 2023). Empirically, we find that the pairs of latents cluster into two distinct modes: high-similarity **hits** where a latent has been matched to a close counterpart, and low-similarity **misses** where the latents are relatively unrelated (Figure 1). Qualitatively, hits have a strong tendency to occur in semantically similar contexts and share similar explanations, while misses are usually semantically unrelated.

This bimodal distribution allows us to analyze the *fraction* of features that are shared between two SAEs. For the largest model we tested, Llama 3 8B (Dubey et al., 2024), **only 30% of features are shared** across both seeds. We find that smaller models, and smaller SAEs trained on the same model, tend to have higher fractions of shared features. We also apply the automated interpretability pipeline of Paulo et al. (2024) to compare the interpretability of hits and misses. We find that misses are often quite interpretable, so that an individual SAE training run is likely missing out on a number of interpretable features.

## 2 RELATED WORK

Recent work has found that SAE features are not *atomic*, in the sense that a "meta SAE" can decompose them into more specific features (Leask et al., 2025). Relatedly, a feature in a small SAE may be replaced by multiple, more specific features in a larger SAE. In some cases, a more general feature like *starts with the letter L* appears alongside a specific feature like *the token "lion"*, which may prevent the general feature from being active in contexts where intuitively, both the general and the specific feature apply (Chanin et al., 2024). In light of these phenomena, some have questioned whether the "flat" design of standard SAEs can accommodate the hierarchical structure inherent to human concepts (Ayonrinde et al., 2024). While sparse autoencoders presuppose that neural networks use linear representations, some research suggests that irreducibly nonlinear features also exist (Engels et al., 2024). If this is true, SAEs trained with different random initializations might converge to different ways of "linearizing" the nonlinear features in activation space.

Previous work had found that ReLU SAEs trained with an L1 sparsity penalty were stable under different seeds (Leask et al., 2025; Braun et al., 2024). By contrast, Marks et al. (2024) found that TopK SAEs could be improved by training two different seeds and forcing them to be "aligned," suggesting that they may not be sufficiently aligned by default. A recent benchmark of feature splitting showed a convergent result, where JumpReLU and TopK latents had a higher feature splitting rate than ReLU SAEs (Karvonen et al., 2024).

Concurrently with our work, Balagansky et al. (2024) use the Hungarian algorithm to align features from SAEs trained on adjacent layers of the Gemma 2 models (Lieberum et al., 2024). However, we learned in personal communication with the DeepMind interpretability team that the same random seed was used to initialize every SAE in the Gemmascope collection, so Balagansky et al. (2024)'s positive results are likely dependent on this hyperparameter choice.

## 3 METHODS

We trained several SAEs on the sixth MLP of Pythia 160M (Biderman et al., 2023) with $2^{15}$ latents over the first 8B tokens of its own training corpus, the Pile (Gao et al., 2020), using the `sparsify` library (Belrose, 2024). We use different random seeds for initialization, but SAEs see exactly the same data in the same order. The SAEs we trained have the functional form

$$\hat{x} = \mathbf{D}\,\mathrm{TopK}(\mathbf{E}x + \mathbf{e}) + \mathbf{d} \tag{1}$$

where $x$ is the output of the MLP, $\hat{x}$ is the reconstructed output of the MLP, $\mathbf{E}$ and $\mathbf{e}$ are the encoder weight and bias, and $\mathbf{D}$ and $\mathbf{d}$ are the decoder weight and bias. SAEs are trained to minimize the mean squared error $||x - \hat{x}||_2^2$ between the model's output $\hat{x}$ and the target module output $x$. The SAEs are trained using the Adam optimizer (Kingma & Ba, 2015), with sequence length of 2049, and a batch size of 32 sequences. We also trained SAEs in SmolLM (Allal et al., 2024), GPT-2 (Radford et al., 2019) and Llama 3.1 8B (Dubey et al., 2024). The GPT-2 SAEs where trained on OpenWebText (Gokaslan et al., 2019), the LLama 3.1 SAEs on RedPajama V2 (Computer, 2023), and the SmolLM SAEs on Fineweb-edu Lozhkov et al. (2024). All SAEs had a different number of latents, but all had a ratio of 36 between the number of latents and the dimension of the input.

The exact cost function that we feed into the Hungarian algorithm is a hyperparameter in our method. In particular, we can try to maximize the average cosine similarity of the encoder vectors or the decoder vectors. While the decoder is initialized using the transpose of the encoder, the two matrices can diverge during SAE training. To err on the side of conservatism, we compute one matching using the encoder and another matching using the decoder. **We consider a feature to be shared if its latents are paired together in both the encoder and decoder matchings,** *and* **in both of these matchings they have a cosine similarity of** 0.7 **or greater**. If a latent does not refer to a shared feature, we will call it **unpaired**.

More specifically, we use SciPy's (Virtanen et al., 2020) function `linear_sum_assignment` to find the decoder permutation $\mathbf{P}_{dec}$ that maximizes $\text{tr}(\mathbf{P}_{dec}^T \mathbf{D}_1^T \mathbf{D}_2)$ and the encoder permutation $\mathbf{P}_{enc}$ that maximizes $\text{tr}(\mathbf{P}_{enc}^T \mathbf{E}_1 \mathbf{E}_2^T)$, where the columns of $\mathbf{D}_1, \mathbf{D}_2$ and the rows of $\mathbf{E}_1, \mathbf{E}_2$ have been standardized to unit norm beforehand. Shared features correspond to pairs of latents $(i, j)$ where the $i^{\text{th}}$ columns in $\mathbf{P}_{dec}$ and $\mathbf{P}_{enc}$ (or equivalently, the $j^{\text{th}}$ rows) are equal. Additionally, the matched cosine similarities $(\mathbf{P}_{dec}^T \mathbf{D}_1^T \mathbf{D}_2)_{ii}$ and $(\mathbf{P}_{enc}^T \mathbf{E}_1^T \mathbf{E}_2)_{jj}$ are required to be greater than 0.7. By this definition, **only 42% of latents are shared** across our two independently trained SAEs. Interestingly, the fraction of shared features is essentially unchanged if we were to use maximum cosine similarity (see below) in lieu of the Hungarian algorithm to label features as shared (Figure A2).

**Maximum cosine similarity.** Prior work has measured the similarity of independently trained SAEs using the *mean maximum cosine similarity* (Braun et al., 2024, Figure 3b). Specifically, for each latent in the first SAE we find the maximum cosine similarity between itself and all latents in the second SAE. The average of these maxima is the overall similarity score. This metric is simple, but it has the downside that it does not yield a bijective matching: many latents in the first SAE may be mapped to one latent in the second SAE. We compare the matched cosine similarity produced by the Hungarian algorithm to the maximum cosine similarity for each latent in Figure A3, observing that while for some latents the max cosine similarity is higher than the matched cosine similarity, the vast majority have the same value for both metrics, suggesting that the Hungarian algorithm has chosen to match most latents with their nearest neighbors. While we think the Hungarian matching approach is more principled and use it in the rest of this paper, we do find that empirically the difference between these two approaches is small.

**Interpretability** We use the automated interpretability pipeline released by Paulo et al. (2024) to generate explanations and scores for the SAE latents. For each latent, representative samples of its activations are sampled and shown to an LLM, in our case Llama 3.1 70b Instruct (Dubey et al., 2024), which is told to generate a succinct explanation that summarizes the activations. The LLM is shown 40 examples sampled from the whole activation distribution.

After explanations are generated for each latent, they are scored. We also use the pipeline from Paulo et al. (2024) for this process. We use both fuzzing and detection to score the latents. To compute the detection score, Llama 3.1 70b Instruct is given the explanation of the latent and a set of examples. The LLM then has to decide which examples activate the latent and which don't using the explanation that was given. At the end the detection score of the latent is given by the balanced accuracy, which in our case reduces to accuracy because we use the same number of activating and non activating examples. The fuzzing score is computed with a similar protocol, but LLM is instead tasked to identify if a given highlighted token is active given the explanation.

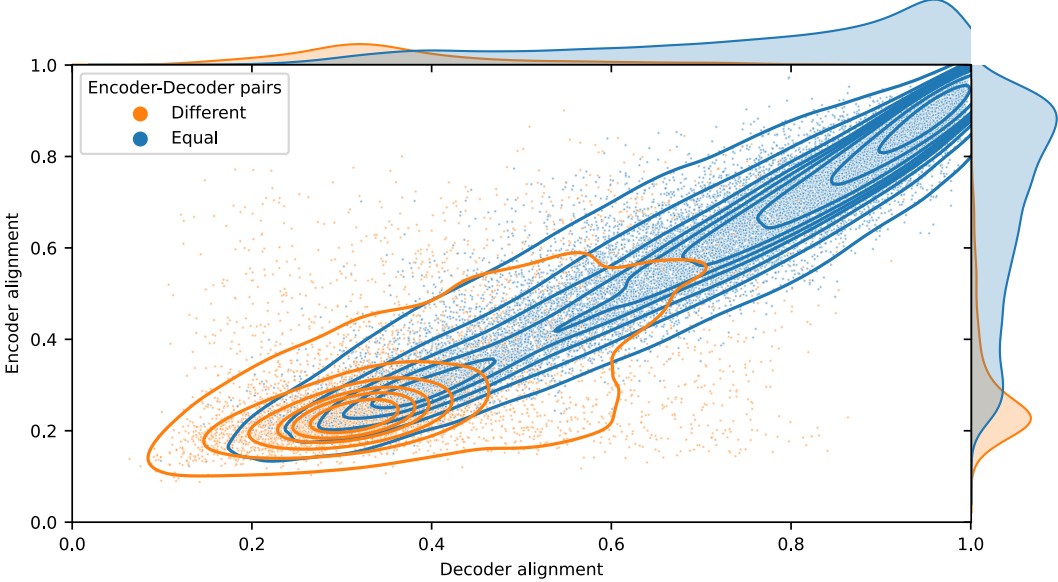

Figure 1: **Cosine similarities of latents from SAE 1 with their counterparts in SAE 2.** Both SAEs have 32K latents, and are trained on the sixth MLP of Pythia 160M. Contour lines are regions of equal density according to kernel density estimation. We color each SAE 1 latent depending on whether the encoder-based and decoder-based matchings agree on which counterpart it should get.

## 4 RESULTS

On this pair, we find that the distribution of matched cosine similarities has two modes: high-similarity hits and low-similarity misses (Figure 1). Overall, cosine similarities for encoder and decoder vectors are strongly correlated. We observe that in cases where the encoder and decoder matchings disagree (colored in orange), the cosine similarity is usually low for both matchings, whereas similarities are higher when the encoder and decoder matchings agree (colored in blue).

### 4.1 ASYMPTOTIC TREND

We now consider seven more SAEs with the same data order, but with seeds different from the first two, yielding a total of nine independently trained SAEs. We first run the Hungarian algorithm $\binom{9}{2} = 36$ times, one for each pair of SAEs. Then for each integer $k$ from 2 to 9, we iterate over all $\binom{9}{k}$ combinations of SAEs of size $k$, and for each combination, we run the following experiment $k$ different times, each time using a different SAE as the "base SAE." We use each of the $k - 1$ matchings of the base SAE with a different SAE within this combination to compute a binary mask classifying each latent as a hit or a miss, using the definition from Section 3.

We say that a latent is "only in the base SAE" if it is an miss according to all $k - 1$ of these binary masks. Then, with respect to a given base SAE, we compute the proportion of all latents that are only in the base. Finally, we average the proportions generated by running this experiment $k \times \binom{9}{k}$ times, one for each combination of $k$ SAEs and each possible base SAE in each combination. When $k = 9$, we find that number of latents found in only one SAE is reduced to about 35% . The results of these experiments are plotted in Figure A1.

The number of latents found in only one SAE decreases slowly as the number of seeds increases Figure A1. Our results indicate that when training a small number of SAEs, a certain number of latents will never find a counterpart – we found that a power law with an offset term fits the data significantly better than one without the offset.

To generate Figure 2, we fix Seed 1 as the base SAE, and color latents based on the number of matchings (out of the $9 - 1 = 8$ matchings involving Seed 1) in which they find a counterpart. We find that the latents that most frequently fire in the first SAE are the ones that have a counterpart in all

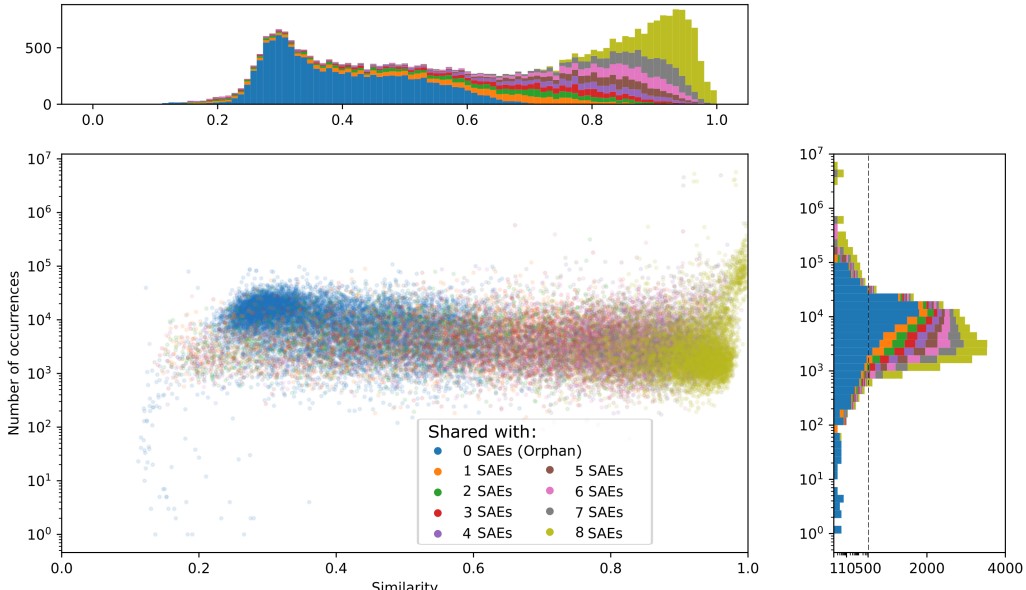

Figure 2: **Latent similarity vs. firing frequency.** We plot the cosine similarity between matched latents, vs. how often the latent fires in the base SAE. The similarity of each latent is averaged over all the matched latents of different seeds. The histograms in this figure are stacked, and the histogram of number of occurrences has a log-scale from 0 to 500, to highlight the few latents that rarely fire or that fire frequently, and a linear-scale from 500 to 4000. Latent occurrences were collected over 10M tokens of the Pile, the same dataset that the SAEs were trained on.

eight SAEs, and that the ones that most infrequently fire are the ones that do not have a counterpart in any other SAE, see Figure 2. Interestingly, a significant number of misses have a higher firing rate on average than latents with counterparts in all SAEs. In fact, as the average alignment between latents increases, the firing frequency seems to decrease. This non monotonic relationship is better observed in Figure A6.

## 4.2 ARE THE UNPAIRED LATENTS INTERPRETABLE?

In this section, we generate explanations for all latents of two seeds of a $2^{15}$ latent SAE and score them using detection and fuzzing scoring (Paulo et al., 2024), evaluating the explanation over 100 active sequences and 100 non-active sequences. The average score of the explanations of the 32K SAEs is 0.72, with only 25% of explanations having a score lower than 0.62, and only 25% having a score better than 0.8.

Plotting the distributions of scores conditioned on the number of SAEs that "shared" that latent reveals that features shared across a higher number of SAE seeds have on average higher interpretability scores. In spite of this, a significant fraction of latents found only on one SAE, have high scoring latents. Plotting the scores of the latents of the two seeds mentioned above, we find that the most of the latents that have low similarity have either a low or an average score, see Figure 3 left. Some latents have a average cosine similarity $< 0.7$ and high scores, reinforcing the observation that some interpretable latents can be missing from any given seed, see Table 1 for some examples.

## 4.3 ABLATIONS

We performed several ablation studies to investigate how our results depend on the hyperparameters used to train the SAE, including the number of active latents $k$, the total number of latents, the number of tokens used for training, and the SAE architecture (TopK, Gated, or ReLU).

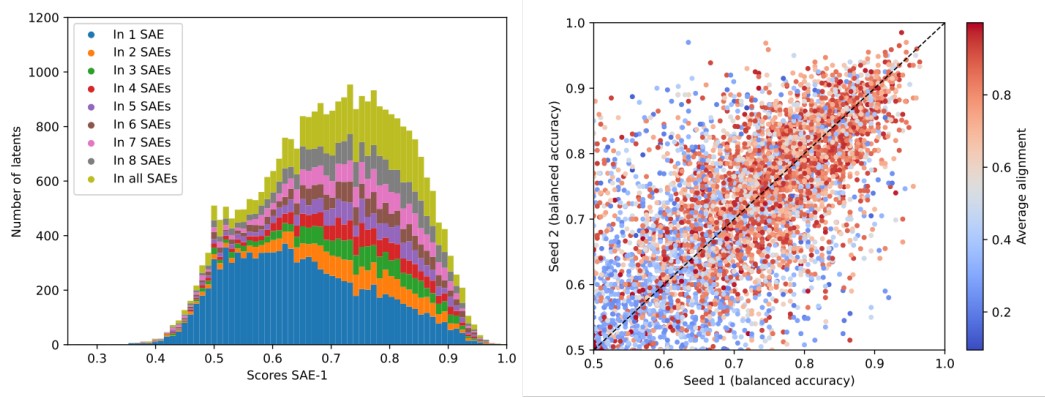

Figure 3: **Interpretability of unpaired latents.** Distribution of scores of different latent explanation conditioned on the number of SAEs that latent can be found on. On the right we compare the scores of 5k explanations of matched latents of different SAE seeds. We see that most of the latents that have low alignment either have a low score or have a higher score in one of the SAEs than in the other.

| Alignment | Seed 1 | Seed 2 |
|---|---|---|
| 0.10 | Abbreviated country name in United States Supreme Court case citations. (0.865) | A single character or a small group of characters embedded within a larger word, often in a non-English language context (...). (0.56) |
| 0.27 | Definite articles and other words commonly used in formal and legal language, such as disclaimers, licensing terms, and court documents. (0.91) | Punctuation marks or short words connecting or separating clauses, (...) and sometimes serving as conjunctions or prepositions. (0.46) |
| 0.44 | Abbreviated geographical or institutional references, usually in the context of legal citations. (0.85) | Punctuation marks. (0.49) |
| 0.75 | Percent symbols marking numerical values representing proportions or rates. (0.95) | A percentage symbol denoting the proportion of a quantity, (...) and usually in the form of a numerical value followed by the symbol. (0.97) |
| 0.97 | Adverbs that express frequency, such as 'often', 'sometimes', (...), used to indicate the occurrence or tendency of an event or action. (0.94) | Adverbs indicating frequency, such as 'often', 'frequently', (...), are used to describe the regularity or likelihood of an event or situation. (0.99) |

Table 1: **Unpaired latents can have high scoring explanations.** We selected explanations of pairs of latents shown in Fig. 3. Each explanation is shown alongside its detection score (Paulo et al., 2024), a number in $[0, 1]$ measuring explanation quality, in parentheses. We select latents from 5 bins of alignment by maximizing the score of both explanations if the cosine similarity between the latents is $> 0.7$ and by maximizing the score of the explanation on seed 1 and minimizing the score on seed 2 of the cosine similarity is $< 0.7$. Ellipsis added to some explanations for brevity. This choice was made to capture latents that had good explanations in seed 1 but were not matched in seed 2. The latent pairs are $(12314, 6024)$, $(21463, 3361)$, $(5888, 6649)$, $(14931, 5456)$ and $(1817, 66)$.

We find that increasing the number of SAE latents, all else being equal, decreases the overlap between different seeds, see Figure 4. Increasing the number of active latents, by increasing the $k$ for TopK SAEs, also decreases the overlap, while the training time increases the overlap between latents. These results seem to indicate that what the seed dependence is not mainly due to feature absorption, as absorption increases when sparsity is decreased (Karvonen et al., 2024), and the model is trained for longer, while it does increase when the number of latents increases. We have found

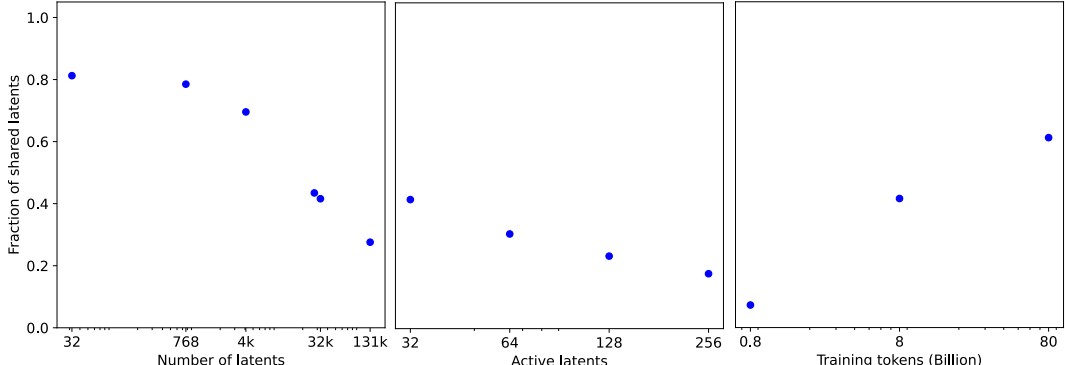

Figure 4: **Dependence of overlap of a Pythia-160M SAE on size, number of active latents and training time**. On the left we see that the fraction of shared latents decreases with the increase of the number of latents. Middle shows that increasing the number of active latents, by increasing the value of $k$ for the TopK activation function, also decreases the overlap. On the right, training time increases the alignment of different SAE seeds. Unless otherwise indicated, each SAE has $2^{15}$ latents and was trained on the output of the sixth layer MLP of Pythia 160M, on the first 8B tokens of its training corpus, the Pile.

no evidence of feature absorption on the MLP SAEs we trained, but that may be due to the fact that the current metric is not tuned to find absorption on MLP SAEs, as it was mostly used on residual stream ones.

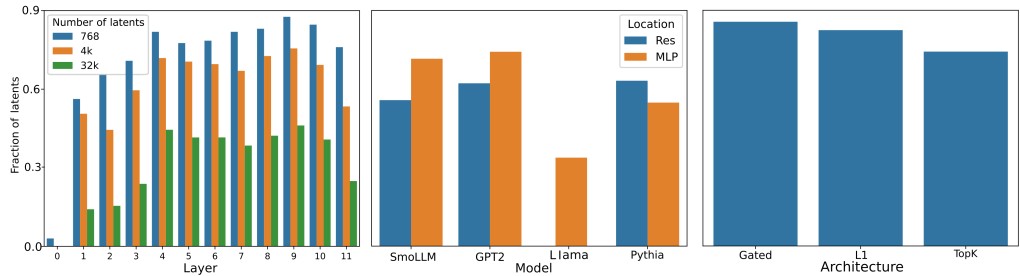

Figure 5: **Dependence of overlap on SAE hyperparameters.** On the right we see the how the fraction of shared features for a Pythia-160M SAE depends on the layer and on the number of latents. In the middle we compare SAEs with the same expansion factor, 36, trained on different models and positions. On the right we compare SAEs trained on GPT2 using different activation functions and architectures.

The overlap between different seeds remains almost constant across the middle layers of the model, being lower for the earlier layers and the last layer (Figure 5). On SmolLM (Allal et al., 2024) and GPT2 the MLP latents have more overlap between seeds than the residual stream ones, but the same is not true for Pythia. Previous work had found that a large number of latents ($> 90\%$) where shared between GPT2 seeds (Leask et al., 2025; Braun et al., 2024), although those numbers where measured for SAEs with smaller numbers of latents than ours, and using a different architecture (ReLU instead of TopK). Indeed we find that standard and Gated SAEs (Rajamanoharan et al., 2024) trained with L1 loss have a larger overlap between latents. We also find that the overlap is much lower for the Llama 8B TopK SAEs, which have more latents but the same expansion factor.

In Figure A4 we compare the matched cosine similarity produced by the Hungarian algorithm to the maximum cosine similarity, showing that these are strongly correlated. This shows that our results are not strongly dependent on the choice of method used to compare latents from independently trained SAEs.

## 5 CONCLUSION

Our results are further evidence for the idea that SAEs do not uncover a "universal" set of features. Different random initializations can lead to different sets of features being found, and SAEs seem to diverge, rather than converge, with increasing scale. We think feature discovery is best viewed as a compositional problem, wherein we look for useful ways of cutting up the activation space into categories, and these categories can themselves be cut up into further categories, hierarchically.

Mathematically, the lack of universality we observe here is due to the nonconvexity of the SAE loss function, which gives rise to many local optima. The TopK activation function is also discontinuous, which may exacerbate the problem of nonconvexity and explain why TopK SAEs suffer from seed instability more than ReLU SAEs. One might have expected a priori, however, that different local optima would have more feature overlap than we found in this study.

## 6 CODE AVAILABILITY.

Code to perform the Hungarian alignment of SAE seeds as well as all the analysis done in this work can be found here, and all the trained SAE checkpoints can be found here.

## AUTHOR CONTRIBUTIONS

Gonçalo Paulo had the idea of investigating the overlap in features learned by SAEs trained with different random seeds, trained the SAEs, and ran all the experiments. Gonçalo wrote the first draft, and Nora Belrose extensively edited the draft into a final version.

## ACKNOWLEDGEMENTS

Gonçalo and Nora are funded by a grant from Open Philanthropy. We thank Coreweave for computing resources.

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

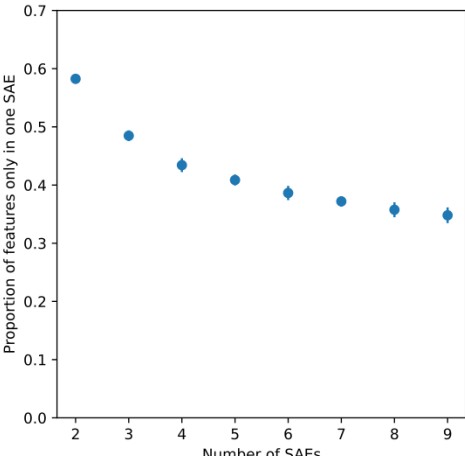

Figure A1: **Dependence of the number of latents found only in the base SAE on the number of seeds.** We consider a latent X in SAE A to be "shared" in SAE B if and only if X is matched to a latent Y in B with which it has cosine similarity greater than 0.7 according to both the encoder and decoder weights. To generate this plot we select a "base" SAE and compute its overlap with all the other seeds, then we average over all different base seeds.

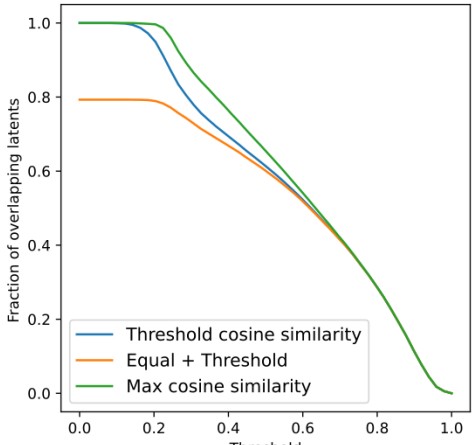

Figure A2: The average alignment of points with equal decoder and encoder indices is 0.72 and of the ones that have different indices is 0.33. On the right, we plot the fraction of latents that are considered shared between 2 SAEs as we control a threshold. We decide to use a threshold of 0.7 on both the encoder and decoder alignment to decide if a latent is shared between two SAEs.

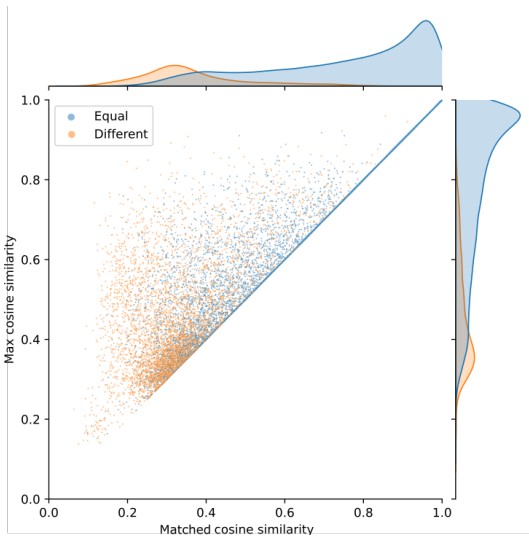

Figure A3: **Cosine similarity of latents when paired with the Hungarian algorithm vs when using max cosine similarity**. The majority of latents that have the same counterpart latent in both the encoder and decoder matchings using the Hungarian algorithm have a similar alignment as if they had been aligned with maximum cosine similarity. The latents which have a higher cosine similarity pair when using max cosine similarity are paired with a latent that already had a pair.

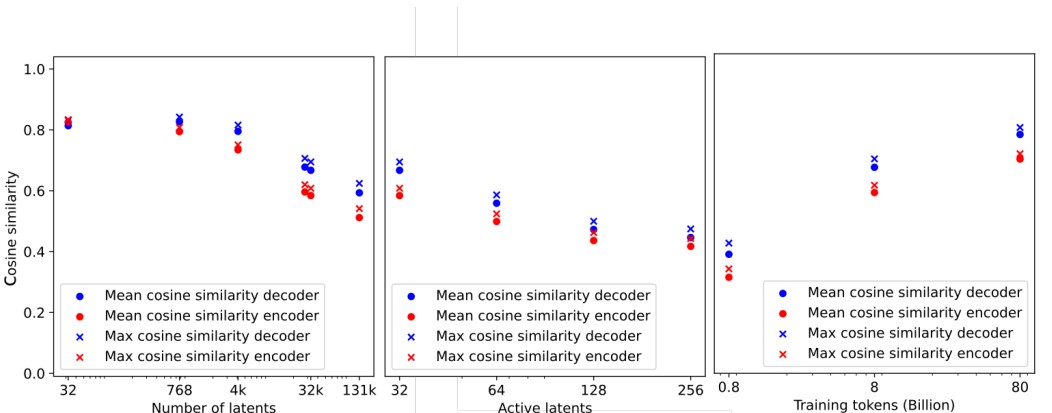

Figure A4: **Dependence of mean matched and mean max cosine sim of a Pythia-160M SAE on different hyperparameters**. On the left we see that the average cosine similarity of latents decreases with the increase of the total number of latents. Middle shows that increasing the number of active latents also decreases the average cosine similarity. On the right, training time increases the average cosine similarity of different SAE seeds. We observe that the mean matched and max cosine similarity have very similar trends, with max cosine similarity being just slightly higher. On panels all panels a 32768 latent SAE was trained on the output MLP of Pythia 160M, for 8B tokens, except when the panel changes one of these conditions.

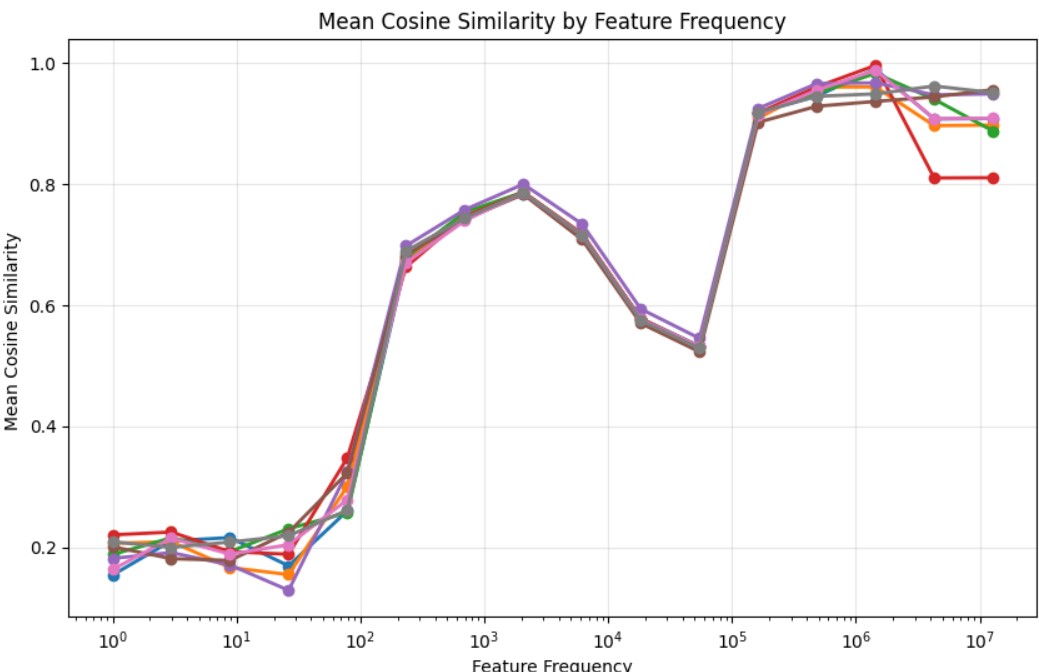

Figure A6: **Frequency dependence of similarity between features**. We find a non monotonic dependence between frequency and cosine similarity. The different colors represent the different seeds. We always use the same SAE to compute the feature frequency.

