# OpenReview forum: "Sparse Autoencoders Trained on the Same Data Learn Different Features"
_ICLR.cc/2026/Conference — ICLR 2026 Poster_

### Official Review · Reviewer_66RK · 2025-10-15

**Soundness:** 3
**Presentation:** 3
**Contribution:** 4
**Rating:** 8
**Confidence:** 3

**Summary:**

This paper engages with the question of wether SAEs learn some fundamental set of features, which is a common theme in the interpretability literature. The paper studies this using feature alignment (an approach from the literature), and runs evals across multiple models, saes, and datasets.
In particular, the paper checks feature alignment across different seeds, and it checks the interpretability of aligned/not features.
It also studies the sensibility of alignment to hyperparameters such as the sae expansion factor and k in topk.

**Strengths:**

This paper is rigorous and works across different families of models, saes, and datasets.
The paper studies a fundamental question to the development of SAEs: wether they learn a universal set of features or not, and it makes a meaningful contribution to the discussion.
The findings are very interesting (and I believe will be interesting to the community), the highlights were:
- features learned across different seeds are significantly different
- the interpretable features were more aligned
- increasing the number of latents reduced alignment, and so does increasing k in topk

This leads to interesting places, for ex:
- maybe the linearly encoded learnable interpretable features only represent a fraction of the total loss (see dark matter sae paper)
- maybe the SAEs have a massive tail of niche features making up a significant portion of the loss, and pick different ones initially due to the seed, but asymptotically we would see all features aligning

**Weaknesses:**

Fundamental:
- the paper could use some human interpretability besides fuzzing, as auto interp can be misleading

Style:
- it is not customary to cite personal communication with another team (about gemma scope in this case) as supporting evidence in an argument
- typo at line 209 "an miss"

**Questions:**

Have you tried experiments on small models with extremely large values of k or the sae expansion factor (to see if the alignment eventually converges)?

Have you tried human interp?

---

> ### Author Response · Authors · 2025-11-25
>
> We thank the reviewer for your helpful comments. We would just like to address the weaknesses you identified in our paper.
>
> > the paper could use some human interpretability besides fuzzing, as auto interp can be misleading
>
> We do not use human interpretability due to time and money constraints, but we do average fuzzing and detection scores in order to increase the robustness of our results.
>
> > typo at line 209 "an miss"
>
> We have now fixed this typo.

---

### Official Review · Reviewer_rLR5 · 2025-10-27

**Soundness:** 3
**Presentation:** 3
**Contribution:** 2
**Rating:** 4
**Confidence:** 4

**Summary:**

This paper studies the stability of SAE features learnt on same model & data but different random seeds. The authors compute a bijective matching of features in two SAEs by maximizing the average cosine similarity of encoder vectors or decoder vectors in Hungarian algorithm. The results show that only 30% of features are shared between two SAEs in Llama 3 8B. Interpretability of non-shared features are statistically worse than shared features.

**Strengths:**

1. The paper is well-written. All experimental settings are clarified in detail. Results are easy to follow.
2. The conclusion they reached, that SAE trained on the same data learn different features, is verified in various models, including SmolLM, GPT-2 and Llama 3.1 8B, respectively on their training set. The Impact of hyperparameter selection is extensively discussed in Section 4.3.
3. Interpretability of shared features versus non-shared features is measured.

**Weaknesses:**

1. Although this paper conducts a wide range of experiments on the phenomenon, the research question is still a bit simple. The main text focuses on the phenomenonology and empirical studies. The authors do not discuss 1) the underlying mechanism behind the instability of features, 2) the impact of instability of features (how the unpaired latents will harm the interpretability ability of SAEs), or 3) the potential method to consistently discover a universal set of features.
2. While a gap is showed in the interpretability of shared features versus non-shared features, the potential causes and its meaning is unclear.

**Questions:**

1. (As in Weaknesses 2) As shown by Wang et al. 2024, shared features may largely be features of low complexity, e.g. single token features. Do the interpretability gap dominants by the complexity gap?
2. What does a single point refer to in Figure 2? Does it refer to a pair of base SAE feature and orther SAE feature, so one base SAE feature will be represented by 8 points in the figure?
3. What is Aligned and Encoder-Decoder equal in Figure 5?
4. The conclusion that L1 is better than TopK contradicts recent work [1] where their experiments show the feature consistency in TopK is significantly higher than standard SAEs. Can you provide any explanation on what possible reason leads to the difference?

[1] Song, X., Muhamed, A., Zheng, Y., Kong, L., Tang, Z., Diab, M., Smith, V. \& Zhang, K. Position: Mechanistic Interpretability Should Prioritize Feature Consistency in SAEs. **Mechanistic Interpretability Workshop At NeurIPS 2025**. (2025), https://openreview.net/forum?id=d9ACURK6bI

---

> ### Author Response · Authors · 2025-11-25
>
> > The authors do not discuss 1) the underlying mechanism behind the instability of features
>
> Thank you for bringing attention to this issue. In our concluding section we do mention the nonconvexity of the loss function as a mechanism for seed instability, and we have now added the following additional text: “The TopK activation function is also discontinuous, which may exacerbate the problem of nonconvexity and explain why TopK SAEs suffer from seed instability more than ReLU SAEs.”
>
> > how the unpaired latents will harm the interpretability ability of SAEs
>
> We argue in our introduction that seed instability makes it impossible to interpret an SAE as “a unique, objective decomposition of a neural network into features.” The uniqueness and objectivity of feature decomposition is presupposed by many agendas in interpretability research.
>
> > the potential method to consistently discover a universal set of features
>
> As we wrote this paper, we considered that the biggest contribution was warning the community of an underappreciated problem with SAEs. At least two different works, developed in parallel to ours, have tried to deal with this problem. Enhancing neural network interpretability with feature-aligned sparse autoencoders, Marks et al. (2024) proposed to improve SAEs by training multiple SAEs in parallel and encouraging their features to be similar. We cite this work in our Related Work section. The paper “Archetypal SAE: Adaptive and Stable Dictionary Learning for Concept Extraction in Large Vision Models” by Fel et al. (2025) also shows that the seed instability of SAEs can be reduced, but not entirely removed, by changing the training objective.
>
> > (As in Weaknesses 2) As shown by Wang et al. 2024, shared features may largely be features of low complexity, e.g. single token features. Do the interpretability gap dominants by the complexity gap?
>
> Could the reviewer provide a complete citation for Wang et al. 2024? We are not sure which paper they are referring to.
>
> > What does a single point refer to in Figure 2? Does it refer to a pair of base SAE feature and orther SAE feature, so one base SAE feature will be represented by 8 points in the figure?
>
> Each point represents a feature in the base SAE. A feature can be an orphan, or it can be shared by up to 8 SAEs.
>
>
> > What is Aligned and Encoder-Decoder equal in Figure 5?
>
> Sorry for the confusion. “Aligned” should have been labeled “Shared,” meaning the fraction of latents that are shared between two independently trained SAEs. “Encoder-Decoder equal” refers to the fraction of latents which are matched to the same counterpart regardless of whether the encoder or decoder vectors are used for matching. We have now simplified the figure to remove the Encoder-Decoder equal information, since it is distracting and somewhat hard to understand.
>
>
> > The conclusion that L1 is better than TopK contradicts recent work
>
> We are unsure why [1] contradicts several other works like ‘Identifying functionally important features with end-to-end sparse dictionary learning’, ‘Sparse autoencoders do not find canonical units of analysis’ and ‘Archetypal SAE: Adaptive and Stable Dictionary Learning for Concept Extraction in Large Vision Models’, which all find standard L1-trained SAEs to be very stable. Figure 3 in the Archetypal SAE paper even compares the two, finding similar results to us. One possible explanation might be that [1]’s L1 SAEs have a much higher number of dead latents, which will tend to decrease seed stability, while in our experiments both TopK and L1 SAEs had almost no dead latents.

---

> > ### Comment · Reviewer_rLR5 · 2025-11-28
> >
> > > Could the reviewer provide a complete citation for Wang et al. 2024? We are not sure which paper they are referring to.
> >
> > Sorry for the confusion. Wang et al. refers to [Towards Universality: Studying Mechanistic Similarity Across Language Model Architectures](https://arxiv.org/abs/2410.06672).
> >
> > Thanks for your clear explanation of all other questions & weaknesses. However, I still think the scope of this work is rather limited, and I hope to see deeper exploration beyond phenomenology. I'll keep my current score.

---

### Official Review · Reviewer_QtLY · 2025-10-30

**Soundness:** 2
**Presentation:** 2
**Contribution:** 3
**Rating:** 8
**Confidence:** 4

**Summary:**

This paper investigates the stability of Sparse Autoencoders (SAEs) by analyzing whether independently trained SAEs (differing only in random seeds) discover the same features. The authors propose using the Hungarian algorithm to compute optimal bijective matchings between latents of different SAEs, allowing for a principled measure of feature correspondence. Experiments across multiple LLMs and SAE variants show that a surprisingly small fraction of SAE features are shared across seeds; overlap depends on SAE size, activation type, training time, and layer position. The paper argues SAEs produce pragmatically useful but non-universal decompositions of activation space.

**Strengths:**

1. Principled and novel matching approach.
The use of the Hungarian algorithm to compute the optimal matching between SAEs trained with different seeds is both reasonable and novel. The paper also validates its effectiveness by showing that matched pairs under the Hungarian algorithm exhibit consistency with cosine similarity, demonstrating that the method produces meaningful alignments.
2. Interesting empirical discoveries from multi-seed experiments.
The authors conduct matching experiments across nine independently trained SAEs and uncover several intriguing empirical patterns, such as the relationship between latent firing rate and matching frequency, and the counterintuitive finding that a significant number of “misses” (unmatched features) have high firing rates. These observations are fresh and thought-provoking for understanding SAE behavior.
3. Interpretability analysis provides practical insight.
By examining the interpretability of unpaired latents, the paper shows that each SAE training run can miss certain high-quality, interpretable features. This finding is practically important and suggests new directions for improving SAE training stability and coverage.
4. Comprehensive ablation study.
The ablation experiments—covering latent size, active latent number, architecture type, and training time—are thorough and enhance the credibility of the results. The consistent trends across these variations further support the paper’s main claims.

**Weaknesses:**

I couldn't identify serious weaknesses. However, I think analysis somehow remains surface-level.

While the experiments are extensive and clearly presented, the overall analysis remains somewhat superficial. The paper mainly reports empirical correlations (e.g., overlap rates, firing frequency trends) without delving into why such phenomena occur or what mechanisms underlie the observed variability across seeds. For instance, there is little attempt to characterize which properties of latents (e.g., frequency, selectivity, activation entropy) predict stability. A deeper or more mechanistic investigation would significantly strengthen the contribution.

**Questions:**

Maybe more theoretical analysis or intuition in SAE training should be explored in future work.

---

> ### Author Response · Authors · 2025-11-25
>
> We thank the reviewer for your helpful remarks. We would just like to reply to one of your comments:
>
> > ...there is little attempt to characterize which properties of latents (e.g., frequency, selectivity, activation entropy) predict stability. A deeper or more mechanistic investigation would significantly strengthen the contribution.
>
> We would like to remind the reviewer that we do plot the relationship between firing frequency and seed stability in Figure 2, but we admit that this plot is not very readable in its current form. We are adding a new figure (Figure A6) which groups features into bins by firing frequency and plots the average seed stability against the average frequency of the bin. It shows an interesting non-monotonic trend. We agree that there is room for more in-depth investigation of other causes of seed stability, but due to time constraints we have left this for future work.

---

### Official Review · Reviewer_XSz3 · 2025-11-01

**Soundness:** 3
**Presentation:** 2
**Contribution:** 2
**Rating:** 2
**Confidence:** 5

**Summary:**

This study studies the stochasticity of which latents are found by an SAE training process. They train many different sets of SAEs which differ only in random seeds, but otherwise vary model, layer, dataset and architecture. They discuss methods for matching features between features and choose the Hungarian algorithm for generating a bijective mapping between features. Their core results show low matching between features, which may be partially mediated by architecture. Smaller SAEs show greater overlap. It is suggested that these results provide further evidence that SAEs do not uncover a "universal set of features".

**Strengths:**

- Quality: In some respects, this investigation is quite thorough, checking how factors including model, layer, architecture or number of latents effect the size of the common set of features found when seeds are varied.
- Clarity: The paper is generally well written and fairly clear. Graphs are of a reasonable quality.

**Weaknesses:**

- Originality: The question is not new, and even very early SAE interpretability work includes examples of training multiple SAEs, varying seeds and inspecting the universality of features found (See Towards Monosemanticity - Bricken et al). The Hungarian algorithm even as applied to this problem is not novel (as mentioned by the paper).
- Significance: Moreover, the idea that SAEs do not find a single basis has been addressed in the literature such as ("Sparse Autoencoders Do Not Find Canonical Units of Analysis" - Leask et al).

How could the work be improved?
- Since the work is sound, the areas for improvement are likely around dealing with the "so what?" element or the "but why?" element.
- For "so what?" - how does the stochasticity / arbitrariness of which features are found effect the usefulness of SAEs? Can multiple SAEs varying only in their seeds be more useful if used in combination?
- For "but why?" - The use of toy models might be useful. What kinds of toy set-ups result in SAE training procedures that are non-deterministic? Possibly Toy models which include underlying features that are nested or overlap such that there is no clear ideal solution. This line of work would be more interesting if it inspires better SAE design.

Additionally, the author could consider Matrioshka SAEs which may be more deterministic.

**Questions:**

- What was the most surprising result in your experiments? Where are you most changing the picture painted by the *recent* literature?
- Did you expect feature absorption to be driving stochasticity in which features are found? What other factors might be driving this?

---

> ### Author Response · Authors · 2025-11-25
>
> We would like to thank the reviewer for their comments. We will reply to the weaknesses presented and then to the questions posed.
>
> > ‘The question is not new, and even very early SAE interpretability work includes examples of training multiple SAEs, varying seeds and inspecting the universality of features found (See Towards Monosemanticity - Bricken et al)’
>
> While it is true that Bricken et al trained more than one SAE, they were actually looking at different seeds of the same transformer trained on a given dataset. By contrast, we examine SAEs trained on the same model, with the same data, with the same batch order. While in their experiments they were expecting to find universal features across different models, we show that different features are found by SAEs trained using different initialization seeds on the same model. Prior work in SAE interpretability has largely disregarded the issue of seed stability. Since this work, several works have been published advocating for training with the effect of the random seed in mind, although citing them would de-anonymize this publication.
>
> > ‘Significance: Moreover, the idea that SAEs do not find a single basis has been addressed in the literature such as ("Sparse Autoencoders Do Not Find Canonical Units of Analysis" - Leask et al).’
>
> Our work is concurrent with that of Leask et al., and in fact we submitted this work to the same conference they did, but unfortunately we were desk rejected due to a technicality. We see their results as complementary to our findings. In theory, it could have turned out that SAEs don’t find canonical units of analysis— because they find units that can be further decomposed— but nevertheless they always find the same units independently of the random seed used for initialization.
>
> > ‘For "so what?" - how does the stochasticity / arbitrariness of which features are found effect the usefulness of SAEs?’
>
> SAEs are often viewed as finding the “true” decomposition of the representations of the models they are trained on. The fact that different seeds yield different decompositions implies that they don’t find a uniquely correct decomposition— at best they find one of many ‘useful’ decompositions. This is a result that extends and complements ‘Sparse Autoencoders Do Not Find Canonical Units of Analysis’. Our results call into question the idea of using SAEs to find the true ‘ontology’ of the underlying model.
>
> > ‘ Can multiple SAEs varying only in their seeds be more useful if used in combination?’
>
> This might be true, although it is currently unclear how to “combine” SAEs trained with different seeds in a useful way, except from merging them. Alternatively, we might be able to use the degree of seed stability as a measure of SAE quality.
>
>
> > ‘For "but why?" - The use of toy models might be useful. What kinds of toy set-ups result in SAE training procedures that are non-deterministic? Possibly Toy models which include underlying features that are nested or overlap such that there is no clear ideal solution. This line of work would be more interesting if it inspires better SAE design.’
>
> Since our work was disseminated as a pre-print, others have proposed different SAE architectures that try to tackle the problem of seed instability, and even investigated toy models as proposed by the reviewer. In this work we were concerned with showing that seed instability does exist in a standard, non-toy case. We think this is a valuable contribution in itself, especially because prior work seemed to suggest that SAEs are relatively seed-stable.
>
>
> > Additionally, the author could consider Matrioshka SAEs which may be more deterministic.
>
> We have trained two Matryoshka SAEs using the dictionary learning library. We modified the code to allow for different initializations but the same data order. We observed that on a 12K feature SAE trained on Pythia 160M, on 0.8B total tokens, the overlap between the two seeds was only 10%, in line with the results we had for SAEs trained for the same amount of tokens. We have added to the appendix figure A5 detailing the results. Interestingly, we find that after matching the features, a significant part of the features remains in the same block. Despite that, and mostly in the last block, features have a very small cosine similarity.

---

> ### Author Response · Authors · 2025-11-25
>
> > What was the most surprising result in your experiments? Where are you most changing the picture painted by the recent literature?
>
> In general, we did not expect SAEs trained on the same data in the same order to converge on different features. Conventional wisdom prior to our work was that the random seed does not matter much. This might be due to the fact that the recent TopK architecture suffers more from this problem than the older ReLU architecture. We also found it surprising that on large models, the number of overlapping features was very low, and that requires a very large number of training tokens to close the gap. We also showed that many interpretable features exist which only appear in one random seed.
>
> > Did you expect feature absorption to be driving stochasticity in which features are found? What other factors might be driving this?
>
> We initially thought feature absorption might be able to explain our results. It seemed like feature splitting would lead to low overlap between features and that the exact splitting could depend on the initial seed. We were surprised when we did not find correlation between the number of split features detected and the amount of feature overlap across seeds. We are unsure if this is due to the fact that feature splitting is still poorly understood (and so are the metrics used to measure it), or if it means that there is a deeper underlying cause.

---

### Comment · Area_Chair_ULRs · 2025-11-25
**Please discuss**

This paper has a wide range of scores. I would love to see the reviewers engage with each other, as well as the author response, and see if they come closer to a consensus.  Have the rebuttals addressed your concerns or clarified anything?

---

### Author Response · Authors · 2025-12-02
**Unified Response to Reviewers**

We thank all reviewers for their thoughtful and constructive feedback. We appreciate the time taken to review our work and are encouraged by the general agreement on the soundness of our experiments.

We addressed concerns related to scope and novelty, raised by XSz3, rLR5:
- Clarification of Scope: Our work uniquely shows that different features are found by SAEs trained using **different initialization seeds on the exact same model and data order**. This is a more stringent test of feature stability than prior work, which compared features across different models or datasets.
- Complementarity to Concurrent Work: Our results are **complementary** to those of Leask et al. (2025). We show that even if features are decomposable, the specific decomposition found is not unique, challenging the notion of a 'true ontology' of features in the model.
We updated the conclusion to suggest that the nonconvexity of the loss function is likely the primary driver of feature instability, noting that the discontinuous nature of the TopK activation function may exacerbate this problem, leading to greater instability compared to ReLU-based L1 SAEs.

We investigated Matryoshka SAEs, which reviewer XSz3 expected to be more deterministic. We found exactly the same results as for other architectures, showing that this architecture does not resolve the seed instability issue.

Reviewer XSz3 raised an apparent contradiction between Song et al. (2025)’s results and ours. But we have pointed out that instead Song et al. (2025)’s results contradict those found in the broader literature, and we suggest a possible explanation for the discrepancy: it might stem from differences in how we handle dead latents. Our L1 SAEs had very few dead latents, while the L1 SAEs in Song et al. (2025) might have had more, which tends to decrease seed stability.

We have also added a new figure (Figure A6) that bins features by firing frequency and plots the average seed stability, revealing an interesting non-monotonic trend that was not visible in the original Figure 2. We hope this adequately addresses the concern raised by reviewer QtLY that “there is little attempt to characterize which properties of latents (e.g., frequency, selectivity, activation entropy) predict stability.”

---

### Meta-Review · Area_Chair_Thgo · 2026-01-08

**Summary:**

Two reviewers strongly support this work for its rigorous methodology and important implications for interpretability research, while one reviewer argues the question has been addressed in prior work. The critic misunderstands the contribution: the value lies in providing clear, systematic evidence that SAEs do not converge to universal features, which has significant implications for the field even if it was previously suspected. The experimental design across multiple models and architectures makes this the definitive study on SAE stability. Therefore, I recommend acceptance as a Spotlight.

**Reviewer Concerns:**

see summary

**Reviewer Scores:**

Sufficient; the disagreement stems from a fundamental difference in how reviewers value empirical validation of known phenomena, and scores would have remained similar.

---

### Decision · Program_Chairs · 2026-01-26

Accept (Poster)